

# Facial flatness indices: application in orthodontics

Chimène Chalala[1], Maria Saadeh[1,2] and Fouad Ayoub[2]

[1] Department of Orthodontics and Dentofacial Orthopedics, Lebanese University and American University of Beirut, Beirut, Lebanon
[2] Department of Forensic Odontology, Anthropology and Human Identification, Lebanese University, Beirut, Lebanon

## ABSTRACT

Facial flatness indices have been used in anthropology to discern differences among populations. They were evaluated on skulls from around the world.

**Aims:** (1) to evaluate the use of facial flatness indices in orthodontics and (2) to assess their variation among malocclusions, age and sex.

**Materials and Methods:** A total of 322 cone beam computed tomography radiographs were digitized and three facial indices (frontal, simotic and zygomaxillary) along with three transverse distances (fmo1–fmo2, zma1–zma2 and n1–n2) were assessed and compared between different groups.

**Results:** The zygomaxillary index was increased in Class II (32.6 ± 0.42; $p < 0.001$) and decreased in Class III malocclusions (29.4 ± 0.66; $p < 0.001$) compared to Class I (31.18 ± 0.3; $p < 0.001$). The frontal and nasal flatness are not characteristic features of any of the sagittal malocclusions. Facial flatness indices did not differ between males and females and between growing and non-growing patients.

**Conclusion:** The position of subspinale point (A point) forward or backward relative to the zygomaxillary width is a factor of assessment of facial flatness. The zygomaxillary index could be helpful in weighting proportionally the width of the maxilla (expansion) relative to its sagittal position in Class II and Class III malocclusions.

## INTRODUCTION

Facial flatness has often been considered in the assessment and evaluation of disparity among populations (*Debets, 1951*; *Oschinsky, 1962*; *Alekseev, 1979*; *Yamaguchi, 1973*, *1980*; *Bulbeck, 1981*; *Rak, 1986*; *Trinkaus, 1987*; *Gill et al., 1988*; *Pope, 1991*, *1992*; *Ishida, 1992*).

Various measurements have been used to evaluate frontal and facial flatnesses in different populations to compare modern to ancient human beings (*Hanihara, 2000*). *Woo & Morant (1934)* evaluated facial flatness on dry skulls and subsequently, many anthropologists have used their method with minor modifications. *Yamaguchi (1973)* proposed three sets of facial flatness measurements: the frontal index, the simotic index (described by *Woo & Morant (1934)*) and the zygomaxillary index (described by *Alekseev & Debets (1964)*). These indices have been mostly used in anthropological studies

Corresponding author
Chimène Chalala,
chimenechelela@ul.edu.lb

| Table 1 | Definitions of the acronyms and specific terms. |
| --- | --- |
| Acronym | Definition |
| ANB | Cephalometric angular measurement formed by the line connecting A point (subspinale) to nasion and another joining nasion to B point (supramentale). This angle describes the anteroposterior relationship between the mandible and the maxilla |
| Class I | Dental relationship described by *Angle (1899)* in which the mesiobuccal cusp of the upper first molar is aligned with the buccal groove of the mandibular first molar |
| Class II | The mesiobuccal cusp of the upper first molar is anterior to the mesiobuccal groove of the lower first molar |
| Class III | The mesiobuccal cusp of the maxillary first molar lies posteriorly to the mesiobuccal groove of the mandibular first molar |

where all measurements have been performed on dry skulls and fossils in many regions around the world (*Hanihara, 2000*; *Fukase et al., 2012a*; *Ishida & Dodo, 1997*; *Green, 2007*; *Dodo, 1983*), and none on three-dimensional radiographs.

In orthodontics, facial flatness is usually assessed clinically through the visualization of the face and has been mostly related to maxillary hypoplasia (*Naini & Gill, 2008*). With the advent of cephalometrics, the position of the maxilla has served as the only practical measurement for flatness through the relative position of point A (subspinale) to the cranial base (SNA angle) (*Steiner, 1953*; *Downs, 1949*; *McNamara, 1984*; *Jarabak & Fizzel, 1972*; *Ricketts, 1960*, *1961*, *1981*). Therefore, the application of facial flatness indices on three-dimensional craniofacial radiographs would help describe and assess the flatness at different level of the face (frontal, nasal, zygomatic) and would allow a new perception of facial flatness in the orthodontic field.

The purpose of this study was to evaluate the use of facial flatness indices in orthodontics and assess the association between facial flatness indices and age, sex and sagittal malocclusions, measured through three-dimensional radiographic technology.

## MATERIALS AND METHODS

Prior to data collection, the study was approved by the Institutional Review Board of the American University of Beirut (IRB ID: BIO-2018-0065) that waived the need for consent form.

### Study population

The pre-treatment cone beam computerized tomography (CBCT) radiographs of 322 patients (121 females, 201 males; age 16.5 ± 11 years) were selected from the database of initial orthodontic records in a private radiologic center.

Excluded were subjects who had previous or current orthodontic treatment, craniofacial anomalies, or low-quality pre-treatment CBCT.

Females younger than 16 years old and males younger than 18 years were considered as "growing" ($n = 244$), and the remaining as non-growing ($n = 78$).

The sagittal skeletal malocclusion was classified into three groups (Table 1):

Class I ($n = 161$): $0 \leq \text{ANB} \leq 4$;

Class II ($n = 136$): $\text{ANB} > 4$;

Class III ($n = 25$): $\text{ANB} < 0$.

## Radiographic analysis

All CBCTs were digitized by one operator (CC) using the View Box 4 imaging software (dHAL Software, Kifissia, Greece).

Nine points were localized on CBCTs as illustrated in Fig. 1:

1. Frontomalare orbitale right (fmo1) and left (fmo2): defined as the junction of the fronto-zygomatic suture and the orbit rim (*Martin & Saller, 1957*).
2. Nasion (n): defined as the suture between the frontal and nasal bones (*Downs, 1949*).
3. Deepest point on the lateral wall of nasal bone right (n1) and left (n2) (*Woo & Morant, 1934*).
4. Nearest point of the median ridge of the nasal bone (n′) (*Woo & Morant, 1934*).
5. Zygomaxillary anterius right (zma1) and left (zma2): defined as the most inferior point on the zygomaxillary suture (*Martin & Saller, 1957*).
6. Subspinale (ss) or point A: defined as the deepest midline point on the premaxilla between the anterior nasal spine and prosthion (*Downs, 1949*).

After digitization, three facial indices (frontal, simotic and zygomaxillary) were computed as follows (Fig. 2):

- Frontal index of flatness: defined as the percentage of the nasion subtense to the chord between the frontomalaria orbitalia (Fig. 1A).
- Simotic index: defined as the percentage of the minimum subtense of the median ridge of the nasalia to the simotic chord (minimum horizontal breadth of the nasalia) (Fig. 1B).
- Zygomaxillary index of flatness: defined as the percentage of subspinale subtense to the chord between the zygomaxillaria anteriora (Fig. 1C).

The subtenses are obtained by direct measurements of the distance from the summit to the chord.

The smaller the value of these indices, the greater the flatness.

In addition, three transversal measurements were performed to assess the facial width at different levels of the face:

1. fmo1–fmo2: distance between right and left frontomalaria orbitalia, measured to assess the width of the head;
2. zma1–zma2: distance between right and left zygomaxillaria anteriora, measured to assess the midfacial width;
3. n1–n2: distance representing minimum horizontal breadth of the nasalia, measured to assess the nasal width.

## Statistical analysis

All measurements were normally distributed for all the compared groups as assessed by the Shapiro Wilks normality test. A three-way between subjects analysis of variance

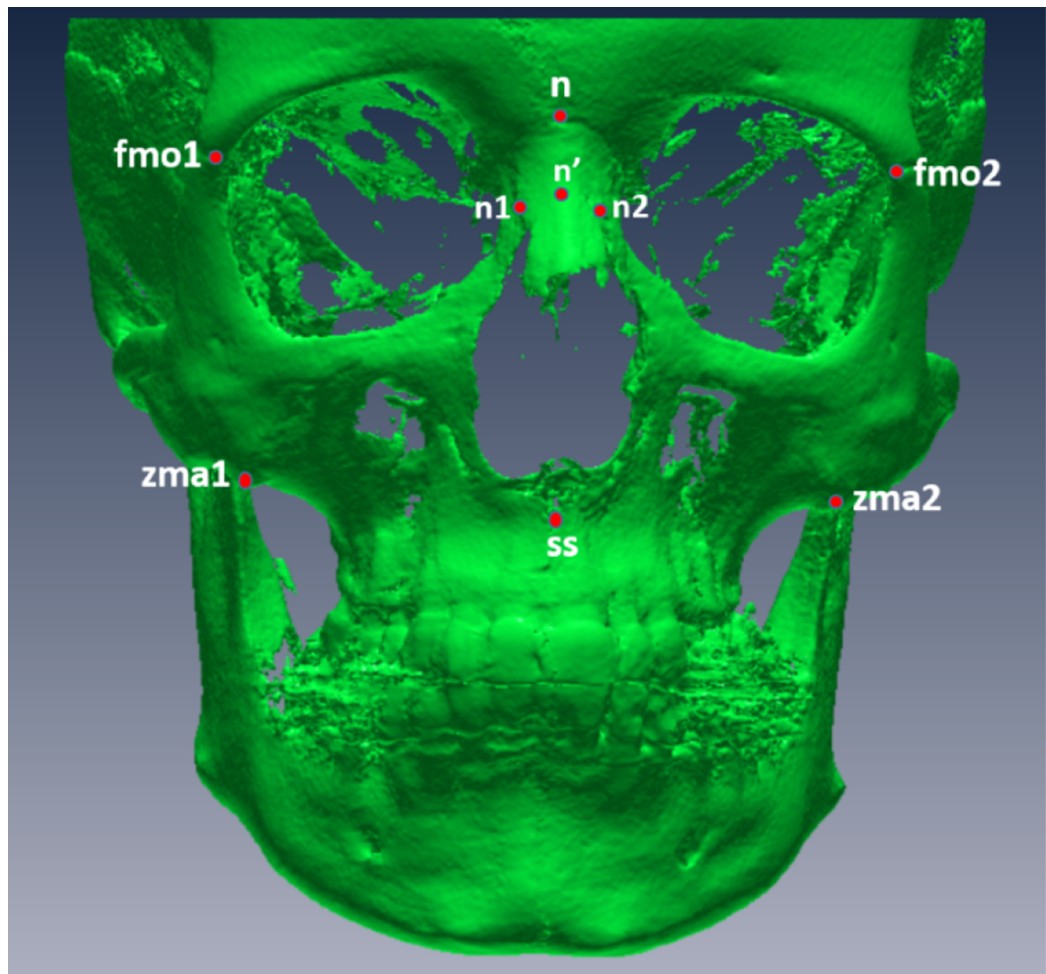

**Figure 1 Digitized points from a frontal view.** fmo1, right frontomalare orbitale; fmo2, left fronto-malare orbitale; n, nasion; n′, nearest point of the median ridge of the nasal bone; n1, right deepest point on the lateral wall of nasal bone; n2, left deepest point on the lateral wall of the nasal bone; zma1, right zygomaxillary anterius; zma2, left zygomaxillary anterius.

(ANOVA) was used to assess the presence of interaction between sex (male, female), growth (growing, adult) and malocclusion (Class I, II, III) on the facial flatness indices and to compare the different groups. When no interaction between the independent variables was detected, a main effect of each variable was reported, followed by multiple comparison post-hoc Tukey tests when comparing the malocclusion groups in case a statistically significant difference was found. The three-way ANOVA was followed by the "simple effects" for the variables that showed a significant interaction between the independent variable.

The three transversal measurements were compared using a two-way ANOVA to check the effect of sex and growth together.

The Pearson product moment correlation coefficient was performed to correlate the widths of the facial structures at different level of the face.

SPSS statistical package was used to perform all tests, at a level of significance of $p \leq 0.05$.

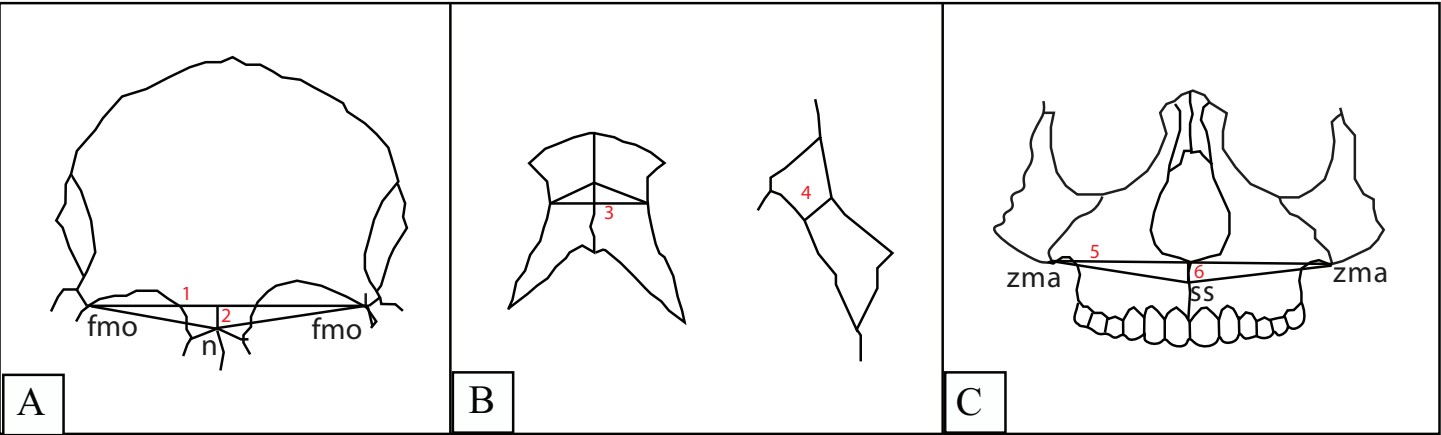

**Figure 2** 1. Frontal chord; 2. Frontal subtense; 3. Simotic chord; 4. Simotic subtense; 5. Zygomaxillary chord; 6. Zygomaxillary subtense. (A) Frontal index: *denominator*: the frontal chord between the frontomalaria orbitalia, *numerator*: the subtense of the nasion from the frontal chord. (B) Simotic index: *denominator*: the simotic chord (the minimum horizontal breadth of the nasal bone), *numerator*: simotic subtense (the minimum distance from the midian ridge of the nasal bone to the simotic chord). (C) Zygomaxillary index: *denominator*: the zygomaxillary chord between the zygomaxillaria anteriora, *numerator*: zygomaxillary subtense (distance from the subspinale to the zygomaxillary chord).

Inter-rater reliability was calculated on all variables of randomly chosen CBCTs ($n = 30$). Intra-class correlation coefficients ranged between 0.82 and 0.99. The lowest coefficient 0.82 was obtained for the distance n1–n2, possibly because of the geometric definition of these nasal landmarks (n1 and n2) in a 3D record.

## RESULTS

### Reliability of the measurements

For all three indices, there was no significant three-way or two-way interaction between any of the variables in their effect on the measurements; thus, the main effects of sex, growth and malocclusion were reported and compared, with the estimated marginal means and their standard errors (Table 2).

There was a significant two-way interaction between sex and growth, on the two distances fmo1–fmo2 ($F = 8.428$, $p = 0.004$) and zma1–zma2 ($F = 4.03$, $p = 0.046$), therefore, the simple main effects of each variable, sex and growth, were reported. For the n1–n2 distance, the interaction was not statistically significant, and subsequently, the main effect was reported (Table 3).

### Sexual dimorphism

None of the three indices displayed a statistically significant difference between males and females, regardless of malocclusion and growth ($p > 0.05$; Table 4).

In both growing and adults subgroups, the fmo1–fmo2 distance was statistically significantly larger in males (93.331 ± 4.582 mm and 100.594 ± 3.501 mm, respectively) compared to females (90.762 ± 3.963 mm and 94.433 ± 3.544 mm, respectively) ($p < 0.001$; Table 3). The interaction between the growth and sex factors is evident with a smaller average difference in growing individuals (2.569 mm) compared to adults (6.161 mm).

**Table 2  Three-way ANOVA for facial indices with sex, growth, and sagittal malocclusion as factors.**

| Variable | df | Frontal index | | Simotic index | | Zygomaxillary index | |
|---|---|---|---|---|---|---|---|
| | | F | p | F | p | F | p |
| Sex | 1 | 0.183 | 0.669 | 1.255 | 0.264 | 0.219 | 0.64 |
| Growth | 1 | 2.17 | 0.142 | 3.462 | 0.064 | 2.641 | 0.105 |
| Malocclusion | 2 | 0.064 | 0.938 | 0.137 | 0.872 | 8.958 | <0.001** |
| Sex * growth | 1 | 0.292 | 0.589 | 0.352 | 0.553 | 0.065 | 0.798 |
| Sex * malocclusion | 2 | 0.422 | 0.656 | 0.264 | 0.768 | 0.022 | 0.978 |
| Growth * malocclusion | 2 | 0.857 | 0.425 | 1.188 | 0.306 | 1.014 | 0.364 |
| Sex * growth * malocclusion | 2 | 0.538 | 0.585 | 0.345 | 0.709 | 0.863 | 0.423 |

**Note:**
** Statistically significant at $p < 0.01$.

**Table 3  Two-way ANOVA results for the transverse measurements with growth and sex as factors.**

| Variable | df | fmo1–fmo2 | | zma1–zma2 | | n1–n2 | |
|---|---|---|---|---|---|---|---|
| | | F | p | F | p | F | p |
| Sex | 1 | 49.765 | <0.001** | 17.367 | <0.001** | 0.649 | 0.421 |
| Growth | 1 | 78.065 | <0.001** | 41.129 | <0.001** | 0.374 | 0.541 |
| Sex * growth | 1 | 8.428 | 0.004** | 4.03 | 0.046* | 0.063 | 0.801 |

**Notes:**
* Statistically significant, $p < 0.05$.
** Statistically significant, $p < 0.01$.

**Table 4  Difference in facial indices and transverse measurements between males and females.**

| | Males (n = 121) | | Females (n = 201) | | Difference (M–F) | | Univariate ANOVA | |
|---|---|---|---|---|---|---|---|---|
| | EMM | SE | EMM | SE | Mean | SE | F | p |
| Frontal index | 18.23 | 0.41 | 18.01 | 0.31 | 0.22 | 0.51 | 0.183 | 0.669 |
| Simotic index | 62.45 | 1.85 | 59.85 | 1.41 | −2.6 | 2.33 | 1.255 | 0.264 |
| Zygomaxillary index | 31.19 | 0.45 | 30.93 | 0.34 | 0.26 | 0.56 | 0.219 | 0.64 |
| fmo1–fmo2 | 94.351 | 0.414 | 91.876 | 0.321 | 2.475 | 0.524 | 22.306 | <0.001** |
| Growing | 93.331 | 4.582 | 90.762 | 3.963 | 2.569 | 0.528 | 23.645 | <0.001** |
| Non growing | 100.594 | 3.501 | 94.433 | 3.544 | 6.161 | 0.119 | 30.311 | <0.001** |
| zma1–zma2 | 83.964 | 0.513 | 82.289 | 0.412 | 1.675 | 0.673 | 6.198 | 0.013* |
| Growing | 82.976 | 6.44 | 81.173 | 5.073 | 1.803 | 0.713 | 6.401 | 0.012* |
| Non growing | 90.006 | 3.648 | 84.851 | 5.125 | 5.155 | 1.51 | 11.656 | 0.001** |
| n1–n2 | 10.519 | 0.256 | 10.279 | 0.15 | 0.239 | 0.297 | 0.649 | 0.421 |

**Notes:**
EMM, estimated marginal means; SE, standard error.
* Statistically significant, $p < 0.05$.
** Statistically significant, $p < 0.01$.

The same trend was found for the zma1–zma2 distance, with a significant difference of 1.803 ± 0.713 mm between growing males and females ($F = 6.401$, $p = 0.012$), and a larger difference of 5.155 ± 1.51 mm in adults ($F = 11.656$, $p = 0.001$).

The n1–n2 distance displayed no significant differences among sex ($F = 0.649$; $p = 0.421$; Table 4).

**Table 5 Difference in facial indices and transverse measurements between growing vs non-growing patients.**

| | Growing (n = 244) | | Adults (n = 78) | | Difference (G–A) | | Univariate ANOVA | |
|---|---|---|---|---|---|---|---|---|
| | EMM | SE | EMM | SE | Mean | SE | F | p |
| Frontal index | 18.49 | 0.24 | 17.74 | 0.45 | 0.75 | 0.51 | 2.17 | 0.142 |
| Simotic index | 58.99 | 1.11 | 63.32 | 2.04 | 4.33 | 2.33 | 3.462 | 0.064 |
| Zygomaxillary index | 31.52 | 0.27 | 30.61 | 0.49 | 0.91 | 0.56 | 2.641 | 0.105 |
| fmo1–fmo2 | 91.857 | 0.282 | 95.776 | 0.498 | −3.919 | 0.572 | 46.896 | <0.001** |
| Males | 93.331 | 4.582 | 100.594 | 3.501 | −7.263 | 1.067 | 46.296 | <0.001** |
| Females | 90.762 | 3.963 | 94.433 | 3.544 | −3.671 | 0.626 | 34.381 | <0.001** |
| zma1–zma2 | 81.941 | 0.361 | 85.974 | 0.639 | −4.033 | 0.734 | 30.201 | <0.001** |
| Males | 82.976 | 6.44 | 90.006 | 3.648 | −7.03 | 1.44 | 23.823 | <0.001** |
| Females | 81.173 | 5.073 | 84.851 | 5.125 | −3.678 | 0.845 | 18.962 | <0.001** |
| n1–n2 | 10.308 | 0.127 | 10.490 | 0.269 | −0.182 | 0.297 | 0.374 | 0.541 |

**Note:**
** Statistically significant, $p < 0.01$.

In the total sample, the largest difference between males and females (2.475 ± 0.524 mm) was found for fmo1–fmo2 ($F = 22.306$; $p < 0.001$) and a difference of 1.675 ± 0.673 mm was observed for zma1–zma2 ($F = 6.198$; $p = 0.013$). Both fmo1–fmo2 and zma1–zma2 distances were significantly larger in males (94.351 ± 0.414 mm and 83.964 ± 0.513 mm, respectively) compared to females (91.876 ± 0.321 and 82.289 ± 0.412; $p < 0.001$ for fmo1–fmo2 and $p = 0.013$ for zma1–zma2; Table 4).

The ratio of the zygomatic width relative to the frontal width is approximately 0.57% more in males compared to females.

## Effect of growth

There was no statistically significant difference in the three flatness indices when assessing the main effect of growth ($p > 0.05$; Table 5).

Both fmo1–fmo2 and zma1–zma2 distances were significantly larger in adults compared to growing individuals, in both males and females subgroups ($p < 0.001$; Table 3). The difference in both distances was found to be approximately twice in males (7.263 ± 1.067 mm for fmo1–fmo2 and 7.03 ± 1.44 mm for zma1–zma2) compared to females (3.671 ± 0.626 mm for fmo1–fmo2 and 3.678 ± 0.845 mm for zma1–zma2) ($p < 0.001$; Table 5).

No difference between growing and non-growing individuals was detected for the n1–n2 distance ($F = 0.374$, $p = 0.541$; Table 5).

Similarly, in the total sample, fmo1–fmo2 and zma1–zma2 distances were significantly larger in adults (95.776 ± 0.498 mm and 85.974 ± 0.639 mm, respectively) compared to growing individuals (91.857 ± 0.282 mm and 81.941 ± 0.361 mm, respectively) and ($p < 0.001$; Table 5). The difference was approximately the same for the frontal and zygomatic width between growing and adults (3.919 ± 0.572 mm for fmo1–fmo2 and 4.033 ± 0.734 mm for zma1–zma2) ($p < 0.001$; Table 5).

**Table 6 Difference in facial indices between sagittal malocclusions (Class I, Class II, and Class III).**

| | Class I ($n$ = 161) | | Class II ($n$ = 136) | | Class III ($n$ = 25) | | Univariate ANOVA | |
| --- | --- | --- | --- | --- | --- | --- | --- | --- |
| | EMM | SE | EMM | SE | Mean | SE | $F$ | $p$ |
| Frontal index | 18.02 | 0.28 | 18.17 | 0.38 | 18.17 | 0.6 | 0.064 | 0.938 |
| Simotic index | 61.32 | 1.26 | 60.35 | 1.74 | 61.78 | 2.75 | 0.137 | 0.872 |
| Zygomaxillary index | 31.18[a] | 0.3 | 32.6[b] | 0.42[c] | 29.4 | 0.66 | 8.958 | <0.001[**] |

Notes:
Alphabetic superscripts denote significantly different column means at $p < 0.05$ (Bonferroni correction).
[**] Statistically significant, $p < 0.01$.

**Table 7 Correlations between the different transversal distances of the face in the total sample and in the males and females subgroups.**

| | zma1–zma2 | | n1–n2 | |
| --- | --- | --- | --- | --- |
| | $r$ | $p$ | $r$ | $p$ |
| Total | | | | |
| fmo1–fmo2 | 0.636 | <0.001[**] | 0.246 | <0.001[**] |
| zma1–zma2 | – | – | 0.077 | 0.168 |
| Males | | | | |
| fmo1–fmo2 | 0.618 | <0.001[**] | 0.18 | 0.048[*] |
| zma1–zma2 | – | – | 0.011 | 0.907 |
| Females | | | | |
| fmo1–fmo2 | 0.636 | <0.001[**] | 0.267 | <0.001[**] |
| zma1–zma2 | – | – | 0.106 | 0.134 |

Notes:
[*] Statistically significant, $p < 0.05$.
[**] Statistically significant, $p < 0.01$.

The ratio of the zygomatic width relative to the frontal width increases with age approximately 0.56% in adults.

### Effect of sagittal malocclusion

When comparing the flatness indices among the three groups of malocclusion (Class I, II and III), there was no statistically significant difference in the frontal ($F = 0.064$, $p = 0.938$) and simotic ($F = 0.137$, $p = 0.872$) indices.

Only the zygomaxillary index displayed a significant difference among malocclusions ($F = 8.958$, $p < 0.001$): it was significantly larger in Class II ($32.6 \pm 0.42$ mm) than Class I ($31.18 \pm 0.3$ mm) followed by Class III ($29.4 \pm 0.66$ mm) (Table 6).

### Correlations

Moderate positive correlations were detected between transverse dimensions at the level of the head (fmo1–fmo2) and the midface (zma1–zma2) in the total sample ($r = 0.636$, $p < 0.001$), and in the males ($r = 0.618$, $p < 0.001$) and females ($r = 0.636$, $p < 0.001$) subsamples. Low positive correlations were found between n1–n2 and fmo1–fmo2 in the total sample ($r = 0.246$, $p < 0.001$) and in males ($r = 0.18$, $p = 0.048$) and females ($r = 0.267$, $p < 0.001$) separately. No significant correlation was found between n1–n2 and zma1–zma2 ($p > 0.05$) (Table 7).

## DISCUSSION

Facial flatness has been evaluated through series of measurements on human cranium throughout the years. Some features related to facial flatness were the subject of interpopulation phylogenetic variations (*Woo & Morant, 1934*; *Weidenreich, 1943*). In 1973, Yamaguchi described three facial indices evaluating flatness at different levels of the face, which have been used only on dry skulls in anthropological studies.

To our knowledge, this is the first study where the flatness measurements are extrapolated and applied on 3D radiographs (CBCTs) to try to induce new interpretations of flatness related to age and sex in orthodontics.

As expected, males in this study presented wider faces than females as the distances between right and left frotomalareorbitale, zygomaxillary anterius and deepest points on the lateral wall of nasalia were increased in males compared to females.

The transversal growth of the face was the most found at the level of the front, less at the level of the midface and the least at the level of the nose.

The present findings are in agreement with the results of previous studies that found sexual dimorphism in some characteristic phenotypes within the facial structures (*Barber, 1995*; *Penton-Voak et al., 2001*; *Perrett et al., 1998*; *Enlow, 1982*), with men tending to have larger facial features than women and even a larger facial-width to-height ratio (*Weston, Friday & Lio, 2007*).

Wider faces were also noted in adults compared to growing individuals of our total sample. The computation of the difference in width of the front and the midface for the growing and adults groups separately was found to be approximately the same (four mm). Additionally, higher correlations were noted in transversal dimensions at the level of the head and the midface but not at the level of the nose. Both distances increase proportionally with age as the ratio of the zygomaxillaryanterius distance over frotomalareorbitale distance remains approximately the same, the difference being only 0.5%. The same trend was found by *Hellman (1935)* who concluded that the transformation of the infant face into that of the adult occurs by increases in size and some changes in proportions.

On the other hand, facial flatness indices did not differ between males and females and between growing and adults. This might be due to the fact that all individuals of the study sample were descended from the same ancestry (*Yamaguchi, 1973*, *1980*; *Rak, 1986*; *Gill et al., 1988*; *Ishida, 1992*; *Woo & Morant, 1934*).

The zygomaxillary index was the only facial flatness index that significantly differed among malocclusion classes: it was significantly decreased in Class III and increased in Class II compared to Class I. As the frontal and simotic index were not different among malocclusions, it could be concluded that the frontal and nasal flatness are not characteristic features of any of the sagittal malocclusions classes.

Therefore, assessment of flatness would be limited to the anterior protrusion of subspinale point with the sole difference that it is weighed relative to the breadth of the zygomatic region and not to the cranial base.

Class II malocclusion has been evaluated through the literature by the amount of maxillary protrusion (subspinale region) relative to the profile (Nasion) (*Steiner, 1953*;

*Downs, 1949*; *McNamara, 1984*). Controversies were found regarding the association between transverse dentoskeletal deficiency of the maxilla and its protrusion in Class II malocclusion. *Franchi & Baccetti (2005)* had found this association to be highly present in Class II subjects mainly with mandibular retrusion. On the contrary, *Vasquez et al. (2009)* had reported no significant transverse deficiency was associated with Class II malocclusion when it is characterized by maxillary skeletal protrusion. Their transversal measurements were expanded to different levels of the face. The outcomes of our study can be helpful to clarify these controversies as the zygomaxillary index was found significantly increased in Class II subjects indicating maxillary protrusion and/or transverse deficient maxilla. Consequently, it will be important to evaluate the zygomaxillary index changes between Class II division 1 and Class II division 2 as the morphologic features are different between the two groups. Specifically, the differential between skeletal projection and dentoalveolar projection in the face might be better understood through the evaluated indices. As orthodontists, we mainly affect the dentoalveolar component. Moreover, it will be valuable to compare this index between Class II malocclusions associated with mandibular retrognathism and those due to maxillary protrusion. In either case, maxillary expansion, if needed, would normalize the zygomaxillary index in Class II subjects and permit to achieve more harmonious and proportional facial features.

In the Class III group, the zygomaxillary index was significantly decreased reflecting more midfacial flatness. According to our study, this flatness at the midface is not noticeable between growing and adults as no significant difference was perceived regarding this index between both groups. Therefore, the flatness of the midface proportionally to the midfacial width would not be worsened with age regardless of mandibular prognathism.

Many systematic reviews and meta-analyses have analyzed the protraction of maxilla using the face mask appliance. They showed a clinically significant improvement in the sagittal relationship between the jaws and forward movement of subspinale point (*Zhang et al., 2015*; *Kim et al., 1999*; *Jager et al., 2001*; *Toffol et al., 2008*). Subsequently, it might be valuable to calculate the zygomaxillary index after maxillary protraction in Class III growing patients to check if any changes in width would have occurred at the level of the zygomaxillary area specially that histological changes of the circumaxillary sutures have been shown in several animal studies (*Dellinger, 1973*; *Jackson, Kokich & Shapiro, 1979*; *Kambara, 1977*; *Nanda, 1978*). In other words, the calculation of the zygomaxillary index before and after maxillary protraction with a face mask would determine if the correction of the Class III malocclusion was achieved by a skeletal or a dental effect and if the midfacial flatness was corrected.

Despite the substantial size of our sample, further research shall help validate our findings, by increasing the sample size within each group. A longitudinal study would better answer the effect of growth on the measured indices, although it is hard to achieve, given the irradiation risk inherent to CBCTs. Future studies would also help investigate more thoroughly the difference in these indices before and after orthopedic Class III correction.

While the zygomaxillary index provides mainly diagnostic instructions, it may indicate treatment limitations. Additional research would hopefully determine such associations following treatment of patients with various indices.

## CONCLUSIONS

1. Facial flatness indices evaluation is important in orthodontics to appraise the harmonization within the different proportions of the face.
2. The zygomaxillary index is decreased in Class III and increased in Class II malocclusions.
3. In Class II malocclusion, the calculation of the zygomaxillary index may be helpful in clarifying the controversies regarding the association between transverse maxillary deficiency and maxillary protrusion.
4. Facial flatness assessment is evaluated according to the position of subspinale point and its projection forward or backward relative to the zygomaxillary width, so proportionally to the width of the midface.

## ACKNOWLEDGEMENTS

The authors acknowledge Dr. Joseph G. Ghafari, Professor and Head of Orthodontics and Dentofacial Orthopedics at the American University of Beirut for his guidance thorough the conduct of this research.

### Funding
The authors received no funding for this work.

### Competing Interests
The authors declare that they have no competing interests.

### Author Contributions
- Chimène Chalala conceived and designed the experiments, performed the experiments, analyzed the data, contributed reagents/materials/analysis tools, prepared figures and/or tables, authored or reviewed drafts of the paper, approved the final draft.
- Maria Saadeh analyzed the data, prepared figures and/or tables, authored or reviewed drafts of the paper, approved the final draft.
- Fouad Ayoub authored or reviewed drafts of the paper, approved the final draft.

### Ethics
The following information was supplied relating to ethical approvals (i.e., approving body and any reference numbers):

The Institutional Review Board of the American University of Beirut granted ethical approval to carry out the study (IRB ID: BIO-2018-0065).

## Data Availability

Raw data is available as a Supplemental File. The pre-treatment cone beam computerized tomography (CBCT) radiographs were selected from the database of initial orthodontic records in a private radiologic center.

## Supplemental Information

Supplemental information for this article can be found online at http://dx.doi.org/10.7717/peerj.6889#supplemental-information.

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
