# Peer review of "Facial flatness indices: application in orthodontics"

_PeerJ, doi:10.7717/peerj.6889_

## Round 0.1 · original submission · Major Revisions

Please be sure to address all of the comments of the reviewers.

Reviewer 1 ·

Basic reporting

The manuscript is clearly written and follows professional reporting standards. All necessary IRB approvals are presented, and raw data is provided and shared according to data sharing policies. The results demonstrate significant findings for orthodontics and studies of human variation, and present novel steps in future studies correlating the two fields of study for future scientific endeavors. The literature cited, background, and discussion are all adequate to support the authors' arguments.

Minor reporting concerns exist, however, which should be addressed:

1) The authors uses the terms 'gender', a socially-based category and identity in lieu of 'sex', a biological distinction between male and female. The binary distinction of sex is used in the present statistical tests, so reporting 'sex' in this manuscript is suggested.The term 'race', while certainly common in the cited anthropological literature, is here used in the discussion (pg 11 ln274) to reflect the relative homogeneity of the present sample population; I would suggest altering the description of the sample population to exclude the historical weight of the term race (e.g. ancestry). See Tishkoff and Kidd (2004) Implications of biogeography and human populations for ‘race’ and medicine. Nature Genetics 36, 11; S21-S27.

2) Minor typos exist throughout (p6 ln80, p11 ln255, p11 ln275 for example).

3) Tables are not consistently formatted: capitalization, cell format, and font vary in some tables.

4) Define terms (e.g., Class I, Class II, Class II) and acronyms (ANB, ENN) for non-orthodontic audience.

Experimental design

The research design is original and significant; the application of long-held anthropological and cephalometric methodologies to contemporary CBCT images provides a framework for testable hypothesis across numerous fields. Statistical methods are valid and pertinent to the present study, and described adequately in the results section. The methods are replicable, with one minor addition:

1) Which landmark had the lower (0.82) ICC (p8 ln160)? Knowing this will help future researchers evaluate Type I and Type II landmarks are most useful for facial flatness indices.

Validity of the findings

The sample size is robust and statistically sounds, and exclusion criteria are valid. No speculation present in the results or discussion, and conclusions are supported by the statistical results.

Additional comments

This is a well-written and impactful manuscript. Minor edits are recommended, many of which pertain to language usage and its relevance/meaning across varying fields, but they do not detract from the conclusions drawn in the study.

Reviewer 2 ·

Basic reporting

The conclusions stated in the abstract do not naturally flow from the stated results. The abstract also lacks statistical evidence of stated results.

I suggest further proofreading of the article. Some phrases are very awkward, e.g., line 66 “modern to old human beings”. This sentence is technically incorrect – “old” human beings would imply individuals who have achieved an advanced age, not fossil hominins.

The summary of the anthropological literature is dated and inaccurate. Facial flatness indices are now rarely used in anthropology. Particularly problematic is the discussion of “mitochondrial Eve” (which was never assumed to be 1 woman, but rather, a small related population). Also, the reason that facial flatness was of interest in paleoanthropology is not clearly articulated. It boils down to the fact that earlier hominins have a greater amount of facial prognathism, and arguments about regional continuity of populations (which were based upon a whole suite of features, facial prognathism being just one). It would be good to clarify the fact that genetic data has essentially resolved the “Out of Africa/Multiregionalism” debate.

The authors alternate between referring to the point of greatest incurvature between ANS and prosthion as “Point A” and “Subspinale”. I recommend choosing 1 term and being consistent. If you choose to use the term “Point A”, please note that most anatomists/anthropologists will not be familiar with this term so it might be useful to initially indicate that this is also known as subspinale.

Please define your technical terms. This paper has been submitted to a journal with a broad scope. A non-orthodontist reader will not know what “ANB” is, for example. Please also define “Class I/II/III” for the reader in more basic (non-measurement-based) terms.

Experimental design

The experimental design appears sound.

Validity of the findings

I am mostly confused about the applicability of the findings, particularly in Class II patients. The authors demonstrate a relationship between the zygomaxillary facial flatness index and skeletal malocclusion, but do not provide any evidence that including this measurement would lead to better/more efficient treatment. Since most practitioners still do not take CBCT scans, what is the added value of taking a scan and collecting this measurement? I think this is especially important given that taking a CBCT scan (may) expose the patient to a greater amount of radiation than a lateral ceph. The authors cite a set of examples in the literature about the transverse relationship of Class II subjects, but it is still unclear how this information would improve treatment. Please clarify.

In the Discussion (lines 273-276) the authors assert that a lack of variation between growing and non-growing patients might be “due to the fact that all individuals of the study sample were ascended from the same race (specie) which confirms their use for the phylogenetic discrimination among populations”. This sentence is problematic. 1) While I assume the authors mean the human race, this is not clear from their wording. (Also, "species" is both the singular and plural form of the word. “Specie” is not correct). 2) The results do not “confirm” the use of these measurements “for the phylogenetic discrimination among populations”. Firstly, what is meant by populations? Different hominin species? Secondly, the authors present no results that indicate that these measurements will discriminate across hominin species (though there is of course other published data on this topic). Please reword this section.

Additional comments

Perhaps a longitudinal study could be accomplished using 3DMD facial images? [This is not a suggestion for the paper, but a more general thought]

Overall, an interesting study applying a new method to (hopefully) improve diagnostic techniques. The applicability of the method to Class II patients should be clarified. Also, the benefits over current methods should be highlighted to address the fact that taking these measurements requires a CBCT scan (still not standard practice for most orthodontists, due to the slightly increased radiation levels compared to a lateral ceph).

---

## Round 0.2 · accepted · Accept

Thank you for following the suggestions of the reviewers.

Reviewer 1 ·

Basic reporting

The suggested edits and recommendations were adequately addressed.

Experimental design

No problems identified- all suggested edits addressed.

Validity of the findings

All edits/rebuttals are sufficient.

Reviewer 2 ·

Basic reporting

The authors have done a nice job of responding to both reviewers’ feedback.
I have noted a few very minor edits:
-Line 274: “ascended from the same ancestry” should be “descended from the same ancestry” (thank you for removing the problematic term “race”)
-Line 150 “Tukey” is misspelled
-Line 284: “Class II malocclusions have” or “Class II malocclusion has”

Experimental design

No issues.

Validity of the findings

No issues.

Additional comments

Thank you for your thorough response to the suggested edits, I believe this has improved the clarity of the article. The definition of key terms in Table 1 should particularly help, given the paper’s goal of integrating anthropological and orthodontic research.